# The Clinical Application of Machine Learning-Based Models for Early Prediction of Hemorrhage in Trauma Intensive Care Units

**DOI:** 10.3390/jpm12111901

**Published:** 2022-11-14

**Authors:** Shih-Wei Lee, His-Chun Kung, Jen-Fu Huang, Chih-Po Hsu, Chia-Cheng Wang, Yu-Tung Wu, Ming-Shien Wen, Chi-Tung Cheng, Chien-Hung Liao

**Affiliations:** 1Department of General Surgery, Chang Gung Memorial Hospital, Taoyuan 333, Taiwan; 2Department of Trauma and Emergency Surgery, Chang Gung Memorial Hospital, Linkou Medical Center, Chang Gung University, Taoyuan 333, Taiwan; 3Department of Cardiology, Chang Gung Memorial Hospital, College of Medicine, Chang Gung University, Taoyuan 333, Taiwan

**Keywords:** machine learning, intensive care unit, traumatic hemorrhage, prediction

## Abstract

Uncontrolled post-traumatic hemorrhage is an important cause of traumatic mortality that can be avoided. This study intends to use machine learning (ML) to build an algorithm based on data collected from an electronic health record (EHR) system to predict the risk of delayed bleeding in trauma patients in the ICU. We enrolled patients with torso trauma in the surgical ICU. Demographic features, clinical presentations, and laboratory data were collected from EHR. The algorithm was designed to predict hemoglobin dropping 6 h before it happened and evaluated the performance with 10-fold cross-validation. We collected 2218 cases from 2008 to 2018 in a trauma center. There were 1036 (46.7%) patients with positive hemorrhage events during their ICU stay. Two machine learning algorithms were used to predict ongoing hemorrhage events. The logistic model tree (LMT) and the random forest algorithm achieved an area under the curve (AUC) of 0.816 and 0.809, respectively. In this study, we presented the ML model using demographics, vital signs, and lab data, promising results in predicting delayed bleeding risk in torso trauma patients. Our study also showed the possibility of an early warning system alerting ICU staff that trauma patients need re-evaluation or further survey.

## 1. Introduction

Uncontrolled post-traumatic hemorrhage is an important cause leading to traumatic mortality that can be prevented [1,2,3,4]. Severe hemorrhage while arriving at the hospital should be detected soon by clinical judgment, image, and laboratory studies. After immediate resuscitation and hemostatic procedures, the patient might suffer from rebleeding from active or occult hemorrhage. Most of these patients were admitted to intensive care units (ICU) for further monitoring and evaluation. Until now, delayed hemorrhage still occurs in trauma ICUs worldwide, and the early detection of hemorrhage can prevent adverse events. Delayed hemorrhage in trauma patients influences multiple systems of the human body. Due to the compensating physiologic function, a significant change in hemodynamics usually represents the late stage of hemorrhage. Meanwhile, the best timing to perform an intervention to stop bleeding can be missed [3,5]. Multi-systemic data evaluation in the ICU has excellent potential for precise, early prediction of hemorrhage events before the bleeding event of the patient, where unstable vital signs or decreased hemoglobin (Hb) are noted by regular follow-up [3,6].

Multiple monitoring approaches and laboratory examinations might be used in intensive care units to achieve better patient care, such as demographic features, physical parameters, laboratory data, and interventional or medical therapy administered to the patients [7]. These clinical data are stored as electronic health records (EHR). Due to the enormous volume of data from the ICU, a high staff-to-patient ratio will be required to analyze and interpret these data to help physicians make clinical decisions [8,9]. However, bias may be inevitable because not all health providers from different fields of expertise operate this system well [10].

Machine learning (ML) is a subfield of artificial intelligence (AI) in which a model learns from examples rather than pre-programmed rules for complex relationships or patterns [6,11,12]. With the availability of healthcare data, enriched data size also empowers the ability and possibility of ML. In sequence, ML became used in various fields of clinical healthcare, from event diagnosis to outcome prediction. Early detection and intervention for patients in the ICU who are fragile to the complication are important for prognosis and the length of the ICU stay [7]. Recently, increasing studies have collected ICU patients’ data to predict clinical events or outcomes [6,7,10,13,14,15,16]. Vezzoli, Marika et al. have applied machine learning for the prediction of in-hospital mortality in the coronavirus disease [17], and except for prediction, Abate, Giulia et al. used machine learning to identify Alzheimer’s disease in preclinical and prodromal stages [18]. As the available data set increased and the technique improved, ML showed earlier advantages in detecting high-risk events earlier [12]. As described above, delayed hemorrhage in trauma patients impacts multiple organ systems. Increasing evidence supports multi-systemic parameters evaluation from the ICU having the potential for precise, early prediction for hemorrhage events that may occur to the patients [3,6]. We currently do not have a proper ML model to predict the delayed hemorrhage event.

In this study, we developed the ML-based algorithm to predict the risk of ongoing hemorrhage events during the ICU stay, and validate the efficacy and efficiency of this algorithm.

## 2. Materials and Methods

### 2.1. Database

We conducted a prospective data collection of the trauma registry (CGTR) since 2008 in Chang Gung Memorial Hospital, Linkou, Taiwan. The summary of demographic data, procedures, hospital course, follow-up, and information regarding the complications of each hospitalized trauma patient was recorded. The Chang Gung Research Database (CGRD) is a multi-institutional electronic medical records (EMR) collection that includes medical documents, laboratory data, vital signs, and nursing records for clinical research across all the hospitals in the Chang Gung Medical Foundation. Both databases can be linked with the same deidentified patient identity.

### 2.2. Data Selection and Inclusion Criteria

The data set was established by acquiring data from CGTR and CGRD from May 2008 to December 2018. The demographic and trauma-related data, including age, sex, date of injury, mechanism of injury, vital signs upon arrival, final diagnosis, associated physiologic and laboratory parameters during the ICU stay, abbreviated injury scale (AIS) score, injury severity score (ISS), and outcomes were collected. Figure 1 demonstrates the data set preparation schema. All the details of the variables are listed in Appendix A.

We enrolled patients diagnosed with torsal trauma with International Classification of Diseases (ICD) diagnosis codes and admitted to the trauma ICU for more than 48 h. The patients with the diagnosis of brain hemorrhage were excluded by the ICD code due to particular concerns and situations of brain hemorrhage that may influence our model building. Detailed information on the ICD codes for including and excluding criteria is listed in Appendix A. The bleeding event was defined as Hb dropping more than 2 mg/dL, as a major bleeding definition according to Schulman, S. et al.’s study [19] from the baseline within 24 h. The study was approved by the Internal Review Board of CGMH No: 202100091B0.

### 2.3. Features Generation and Data Preprocessing

The following variables were used in the development of our model: demographics (ages, gender), GCS scores while arriving at the emergency department (ED) and after leaving the ED, ED evaluation, and clinical events (labeled as trauma team activation or not, abbreviated injury scale (AIS), injury severity score (ISS), intubation or not, cardiopulmonary resuscitation (CPR) or not, received transarterial embolization (TAE) or not), ICU vital signs (blood pressure, heart rate, saturation, respiratory rate, and shock index), and laboratory findings (hematology, including coagulation examination, such as a prothrombin time test and activated partial thromboplastin time, biochemistry, and arterial blood gas).

Each Hb datum during the ICU course can be defined as a bleeding (if Hb drops more than 2 mg/dL in 24 h) or a negative event. Because we are going to build a prediction model to predict the event 6 h before, the variables were collected 48 h to 6 h before the event. Our study’s patients were subgrouped according to bleeding events during the ICU stay. The hemorrhage group is the patients with one or more episodes of bleeding events, and the negative group had no bleeding episodes throughout the ICU course. The negative events outnumbered the positive events. To balance both groups and avoid bias, we prepared the first positive event in the hemorrhage group as the positive sample. For the negative samples, we randomly selected one of the events in the ICU course from each patient in the negative group to balance the two group’s data numbers. Figure 2 shows the current study’s definition of events and data collection.

The vital signs and laboratory data were used as time series variables. Each variable was calculated for the maximum, minimum, median, mean value, frequency of the test, and the difference between each test in the sample to generate more features.

### 2.4. Model Development

Our machine learning models were developed on Waikato Environment for Knowledge Analysis (WEKA, version 3.8.5). The data from two groups (hemorrhage and negative) was input to WEKA. As for attribute selection, we used the gain ratio attribute to evaluate the worth of the attribute by measuring the gain ratio with respect to our result (bleeding event). The variables that attribute more than zero were finally put into our data set to build the model.

After testing several machine learning algorithms, we used the logistic model tree (LMT) and the random forest, which showed better performance with ICU data mining and prediction at our data set, to build our model in order to predict the bleeding event 6 h before it happened. Each model was tested with 10-fold cross-validation to ensure the robustness of the model performance. The data were split into ten equal groups. The first group was left as the test set and others were used to train the model. Then, the second group was left as the test set, and so on. In the end, we will have ten models with performance evaluation. The final result was expressed as the weighted average outcome from these ten models.

### 2.5. Statistical Analysis

Demographic and outcomes data are presented as medians with interquartile ranges. For continuous variables, extensive exploratory data analysis, including the calculation of maximum, minimum, median, and mean values, was conducted; and the frequency of the test and the difference between each test were recorded. The accuracy, sensitivity, specificity, and area under the receiver operating characteristic curve (AUC) with a 95% confidence interval (CI) were used to assess the model performance. Statistical analysis as *t*-tests for calculating *p*-values was performed by Python (scikit-learn Python library). The gain ratio attribute algorithm of WEKA was used to determine feature importance for our model. After model development, SHApley Additive exPlanation (SHAP) was used for the explanation of the feature importance of each model.

The percentage of missing data of each variable in both groups was calculated and we excluded the features with more than 30% missing data. For the rest of the missing data, we imputed it with the mean value of the whole database.

## 3. Results

### 3.1. Demographics and Clinical Characteristics

We collected 2218 trauma patients’ data during the study period (2008 to 2018). The median age was 37 years old (IQR 22–55). Most injured patients were males (1572/2218, 71%). The median ISS score was 20 (IQR 13–29). Forty-four percent of patients arrived with trauma team activation (900/2218). Thirty percent of patients received TAE as bleeding management (770/2218). Twenty-nine patients required CPR (1.8%) and 245 required intubation with mechanical ventilation (245/2218). There were 1036 patients with positive bleeding events during their ICU stay (11%). The remainder of the demographic and clinical characteristics is summarized in Table 1, divided into two groups by having bleeding events or not. Regarding vital signs and laboratory data during the ICU stay, as mentioned before, we calculated and recorded their maximum, minimum, mean, median, and frequency.

### 3.2. Feature Selection Outcome

To clarify the relation between the collected features and our outcome event (Hb dropping), we used the algorithm of the gain ratio attribute of WEKA and the results shown in Table 2, divided into groups of demographic features, vital signs, hematology, and biochemistry.

### 3.3. The Performance of Prediction Models

Two machine learning algorithms were applied in our study. The logistic model tree (LMT) achieves 73.8% precision, 71.3% sensitivity, and 75.9% specificity at the best Youden index of 0.486, with an AUC of 0.816 (95% CI: 0.798–0.834) for the validation set. The random forest algorithm achieves 73.6% precision, 73.6% sensitivity, and 73.7% specificity at the best Youden index of 0.479, with an AUC of 0.809 (95% CI: 0.791–0.827) for the validation set (Table 3). The ROC curve is shown in Figure 3. After model development, SHAP was applied, and the importance of each feature was checked. The temporal change of Hb and hematocrit are critical determining factors. The heart rate and shock index also contribute a lot to the prediction. Among the demographic factors, only ISS, NISS, and angiography affect the prediction. Examples of SHAP plots are shown in Figure 4.

## 4. Discussion

Our study demonstrates that the ML-based algorithm can be a promising tool for evaluating trauma patients in predicting delayed bleeding. It alarms physicians to monitor and take adequate resuscitation and hemostasis for hazardous patients. In this study, a machine learning-based algorithm helps us to identify risky patients with potential delayed hemorrhage events in the next 6 h. Once patients enter the ICU and the data is automatically attributed to the model, the system will warn us if this patient has a potential bleeding risk in the next 6 h. It can help lower the patient’s delayed findings of hemorrhage problems. By early re-evaluation or intervention, the prognosis of the patient can be improved. By this means, it can decrease the dismal outcome that results from bleeding without early detection. The substantial elements of ICU monitoring can assist and compose the algorithm without additional laboratory or examination. The present algorithm can predict the hemorrhage event in the next 6 h with an accuracy of 0.816. There are still limited studies using ML, even deep learning techniques for trauma patients so far. As our research presented, we built an efficient and feasible algorithm to achieve an appropriate level of prediction function. The current study is an exploratory research. Therefore, we only use the ten-fold cross-validation method to validate our result without an external validation data set. Further external validation will be conducted for the best algorithm chosen.

In the current medical environment, numerous data will be generated from patients, which are necessary to be collected to assist health providers in making therapeutic decisions. Especially in the ICU, the most critical patients who need to be monitored the most closely will generate numerous data. How to figure out an efficient and feasible method to predict the prognosis and advanced events is pursued by investigators [20]. For decades, several prediction regression models for evaluating the patient’s bleeding were reported [4,5,14,21,22]. However, additional laboratory examinations or complex calculated formulas were necessary to be applied. Regular blood sample collection and image studies, such as angiography or contrast-enhanced CT, guide our treatment’s direction. In the current study, we developed the algorithm consisting of 56 elements from baseline hemodynamic parameters before entering the ICU, and regular physiologic and laboratory parameters during the ICU stay. No additional work or examination needed to be performed to attain the prediction results. Furthermore, the conventional prediction regression algorithm provided some clues about the linear correlations between clinical characteristics and outcomes [23,24,25]. However, outcomes are usually not linearly correlated with inputs in a real clinical environment. The complicated relationship between the combination of inputs and outcomes makes it difficult to calculate directly. Therefore, the limitation might prevail when applying these regression models to clinical practice. Meanwhile, numerous monitoring features and laboratory examinations might be performed in intensive care units to provide better patient care and facilitate translational research. In addition to the difficulty of developing the algorithm, there are challenges in ICU data usage: [7]. Numerous parameters and records might be generated once patients enter the ICU to define the attributes as ambiguous core elements. Therefore, deciding which attributes are most suitable for the model is often hard. Therefore, attribute selection is complex, and confounding bias cannot be neglected [13,14]. To analyze the immerse data and select proper parameters into prediction models themselves, the advantage of machine learning is that these models assist us in transforming these data into meaningful outcomes [20]; and not only individual parameters, but also a combination of features can integrate. This is the advantage of machine learning for data handling and analysis.

Machine learning showed its greatness in data analytics and complexity regression to predict numerous aspects. Several innovative studies also showed the possibility of ML in handling complicated situations in the intensive care unit. With the supplement of ML, physicians can predict the opportunity and risk of tracheal tube reintubation, the necessity of renal replacement therapy, and sepsis risk and mortality prediction. In the current study, we developed the first ML algorithm to predict delayed hemorrhage in trauma patients, and integrated the ML into ICU to accelerate the data organization, classification, interpretation, analysis, and even data prediction [26].

## 5. Limitation

In this study, we presented the ML algorithm that can predict the delayed or ongoing hemorrhage events in trauma patients once they are admitted to the ICU in the next 6 h with acceptable accuracy. However, there were still some limitations to our study. First, this was a retrospective single-center study, and the nature of the study design means the selective bias cannot be avoided. Second, there were missing data in our database, and we were forced to remove some variables due to many missing data. The reason that we cannot afford too much missing data in variables is to ensure that our outcome relates more to the real clinical situations. We also chose the mean value imputation instead of other complicated imputation methods to avoid over-manipulating the data. In addition, we are trying to build a general analysis system that can be applied to most ICUs. If the variable contains many missing data, it means that this variable may not be collected frequently at the ICU, which is not suitable for our study. Third, as mentioned before, we have not included all the bleeding-related events in our study; thus, we could miss some patients with delayed bleeding, but not expressed as Hb dropping. The blood transfusion event time was also not included in our study due to inaccurate data in the current database. Forth, our study used Hb dropping 2 g per deciliter as a cut point as bleeding events. Besides Hb dropping, there are other events related to hemorrhage. Interventions such as transarterial embolization or exploratory laparotomy/thoracotomy also point to bleeding events that cannot be controlled by conservative treatment. In the databases we used in the current study, we only have raw codes rather than details of procedures. Further work to identify these events and add to our model or build models by a separate event is necessary and helpful for physicians’ decision-making. Furthermore, increasingly innovative tools, such as a pulse-induced contour cardiac output (PiCCO), assisted the intensivist in immediately evaluating the patient’s hemodynamics. By adding these dynamic data, we can expect a more accurate and immediate prediction model to be built in the near future. Because these data are not included in our database, we lack the advanced information to integrate them into this algorithm. Finally, machine learning (ML) is a subfield of AI that focuses on algorithms that allow computers to define a model for complex relationships or patterns from empirical data without being explicitly programmed [10]. How to interpret the result and integrate it into the clinical flow to assist the intensivist in managing their patients is the most critical part, and prospective studies need to be conducted to prove these results.

## 6. Conclusions

In this study, we presented the ML model using demographics, vital signs, and lab data with promising results in predicting delayed bleeding risk in torsal trauma patients. Our study also showed the possibility of an early warning system alerting ICU staff that trauma patients need re-evaluation or further survey.

## Figures and Tables

**Figure 1 jpm-12-01901-f001:**
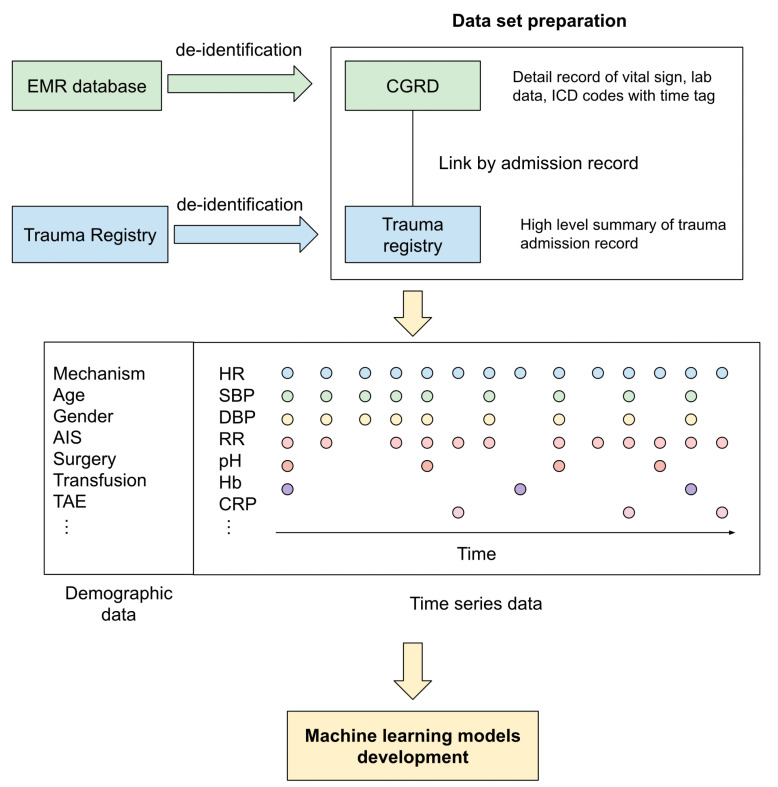
The data set preparation schema in the current study.

**Figure 2 jpm-12-01901-f002:**
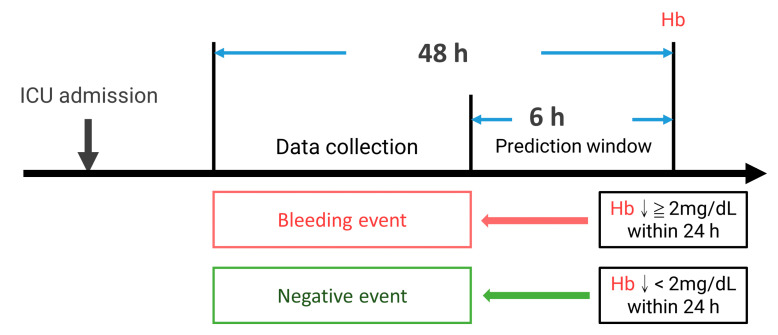
The definition of events and data collection. ↓: drop.

**Figure 3 jpm-12-01901-f003:**
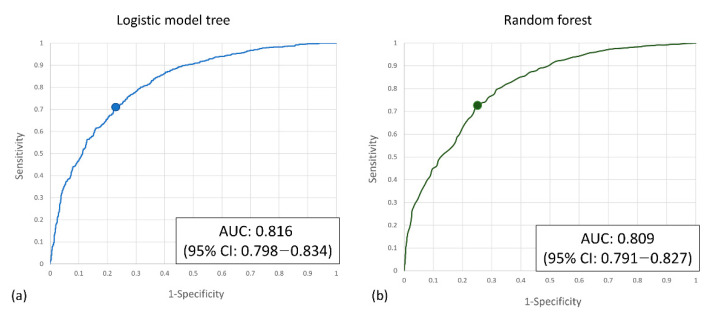
(**a**) The ROC curve of the logistic model tree model and (**b**) the random forest model.

**Figure 4 jpm-12-01901-f004:**
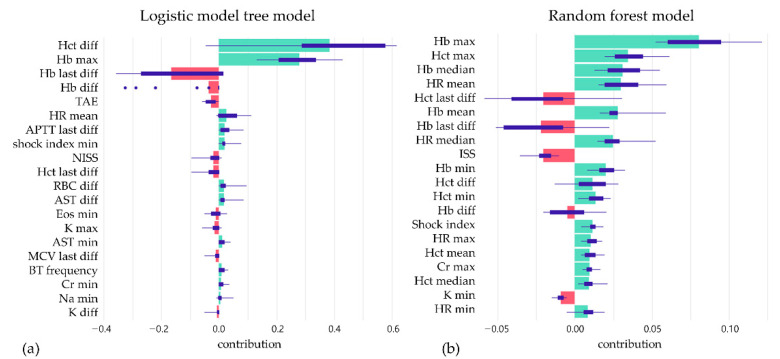
The box plots summarize the distribution of contributions of features in each model by SHAP value; (**a**) logistic model tree model and (**b**) random forest model.

**Table 1 jpm-12-01901-t001:** The demographic feature comparison between each group.

Group	Positive	Negative	*p* Value
No.	1036	1182	
Gender, male, n (%)	782 (75.5)	790 (66.8)	<0.001
Age, median (IQR)	37 (23–54)	37 (225–6)	0.914
ED arrival GCS, median (IQR)	15 (11–15)	15 (13–15)	0.843
ED leave GCS, median (IQR)	15 (11–15)	15 (11–15)	0.805
AIS head, median (IQR)	0 (0–2)	0 (0–1)	0.87
AIS chest, median (IQR)	3 (0–4)	2.5 (0–4)	0.864
AIS abdomen, median (IQR)	2 (0–3)	3 (0–4)	0.868
ISS, median (IQR)	22 (16–29)	20 (13–29)	0.634
NISS, median (IQR)	26 (173–4)	22 (14–29)	0.598
TAE, n (%)	399 (38.5)	371 (31.4)	<0.001
CPR, n (%)	14 (1.4)	15 (1.3)	0.865
Trauma team activation, n (%)	514 (49.6)	466 (39.4)	<0.001
Intubation, n (%)	145 (14.0)	100 (8.5)	<0.001

ED: emergency department, GCS: Glasgow coma score, AIS: abbreviated injury scale, ISS: injury severity score, NISS: new injury severity score, TAE: transarterial embolization, CPR: cardiopulmonary resuscitation.

**Table 2 jpm-12-01901-t002:** The Gain Ratio attribute.

Demographic Feature	Gain Ratio	Hematology	Gain Ratio	Biochemistry	Gain Ratio
NISS	0.0140824	HB last diff	0.060502	ALT max	0.0217857
ED intubation	0.0111678	HB max	0.057131	AST mean	0.0217508
ISS	0.0105278	Hct max	0.055303	AST median	0.0217508
ED trauma blue	0.0076397	Hct last diff	0.049209	AST min	0.0191412
Gender	0.0075227	HB median	0.044343	AST max	0.0171591
ED leave GCS	0.0054404	Hb mean	0.044343	Cre min	0.0157679
TAE	0.0043155	Hct median	0.041516	Cre frequency	0.0153933
ED CPR	0.0000937	Hct mean	0.041516	Cre max	0.0130096
		Hb diff	0.03215	Cre median	0.0126784
Vital signs	Gain Ratio	Hct diff	0.031456	Cre mean	0.0126784
HR median	0.017608	Hct min	0.029003	ALT min	0.0120391
HR mean	0.017608	RBC max	0.028482	ALT median	0.0117981
HR max	0.0142613	Hb min	0.027336	ALT mean	0.0117981
HR min	0.013361	MCV min	0.020603	K diff	0.01081
BT min	0.0107972	RBC mean	0.019859		
SI median	0.0094917	RBC median	0.019859		
SI mean	0.0094917	RBC min	0.016197		
SI min	0.0094543	MCV max	0.016045		
SI max	0.0077304	MCV median	0.015736		
BT frequency	0.0073078	MCV mean	0.015736		
SBP frequency	0.0071975	Platelets frequency	0.013502		

NISS: new injury severity score, ED: emergency department, ISS: injury severity score, GCS: Glasgow coma score, CPR: cardiopulmonary resuscitation, TAE: transarterial embolization, HR: heart rate, BT: body temperature, SI: shock index, SBP: systolic blood pressure, min: minimum values, max: maximum values, diff: difference between maximum and minimum value, last diff: difference between values of the last recorded value and maximum value, RBC: red cell count, MCV: mean corpuscular volume, AST: aspartate amino transferase, ALT: alanine aminotransferase, Cre: creatinine, K: potassium.

**Table 3 jpm-12-01901-t003:** Demonstration of two machine learning algorithm results.

Algorithm		Positive (Pred)	Negative (Ped)	Sensitivity	Specificity	F-Measure	AUC
LMT	Positive (Act)	740	296	71.40%	75.90%	0.718	0.816
	Negative (Act)	285	897				
RF	Positive (Act)	762	274	73.60%	73.70%	0.723	0.809
	Negative (Act)	311	871				

LMT: logistic model tree, RF: random forest, Pred: predicted result, Act: actual result.

## Data Availability

The data were publicly unavailable due to patient privacy and the institute’s data policy. The data can be acquired from the corresponding author upon reasonable request.

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
