# Peer review of "The Clinical Application of Machine Learning-Based Models for Early Prediction of Hemorrhage in Trauma Intensive Care Units"

_jpm, 2022, doi:10.3390/jpm12111901_

Round 1

Reviewer 1 Report

In the manuscript authored by Shih-Wei Lee et al., authors developed a machine learning approach for the Early Prediction of Hemorrhage in Trauma  Intensive Care Units.

The application of AI and machine learning in clinics is a relevant topic that is acquiring higher attention recently. 

Here below I reported my comments and suggestions:

-I suggest authors to better describe the methodology used in the article. 

-In the introduction I suggest authors tofurther describe the recent application of machine learning in clinics also besides trauma, as for example for COVID-19 or Alzheimer's disease prediction to highlight the benefit of applying ML in the hospital to predict mortality and disease progression (doi: 10.2459/JCM.0000000000001329doi: 10.3390/jpm11010014.)

-I suggest including the Yuden index in ROC curves and report the 95 % CI of AUC. 

-Is it possible to calculate the de-long test between the two ROCs to demonstrate if there is a difference between the two algorithms used (Logistic model tree and RF: Random Forest) .

-It is not clear to the reader why the need of two different ML-algorithm. none outperformed the others. Authors should better explain the rationale used in ML-method selection.  

Author Response

Thank you for reviewing our manuscript and giving us constructive suggestions. Below we respond point-by-point to your recommendations.

  1. I suggest authors to better describe the methodology used in the article.

Reply: Thank you for your thoughtful comments. We have added more detail to our method part. Please let us know if there is any other part we need to emphasize, thank you very much!

Page 4

2.3. Features Generation and Data Preprocessing

The following variables were used in the development of our model: demographics (ages, gender), GCS scores while arriving emergency department (ED) and after leaving ED, ED evaluation, and clinical events (labeled as trauma team activation or not, Abbreviated Injury Scale(AIS), injury severity score(ISS), intubation or not, cardiopulmonary resuscitation(CPR) or not, received transarterial embolization(TAE) or not), ICU vital signs (blood pressure, heart rate, saturation, respiratory rate, and shock index) and laboratory findings (hematology, biochemistry, and arterial blood gas).

Each Hb data during the ICU course can be defined as a bleeding (if Hb drops more than 2mg/dL in 24 hours) or a negative event.Because we are going to build a prediction model to predict the event 6 hours before, the variables were collected 48 h to 6 h before the event. Our study’s patients were subgrouped according to bleeding events during the ICU stay. The hemorrhage group is the patients with one or more episodes of bleeding events, and the negative group had no bleeding episodes throughout the ICU course. The negative events outnumbered the positive events. To balance both groups and avoid bias, we prepared the first positive event in the hemorrhage group as the positive sample. For negative samples, we randomly selected one of the events in the ICU course from each patient in the negative group to balance the two group’s data numbers. Figure 2 shows the current study’s definition of events and data collection.

Page 5

2.4. Model Development

Our machine learning models were developed on Waikato Environment for Knowledge Analysis (WEKA, version 3.8.5). Data from two groups (hemorrhage and negative) was input to WEKA. As for attribute selection, we used the GainRatio attribute to evaluate the worth of the attribute by measuring the gain ratio with respect to our result (bleeding event). The variables that attribute more than zero were finally put into our dataset to build the model.

After testing several machine learning algorithms, we used the Logistic model tree (LMT) and the Random Forest, which showed better performance with ICU data mining and prediction at our data set, to build our model to predict the bleeding event 6 hours before it happened. Each model was tested with 10-fold cross-validation to ensure the robustness of the model performance. The data was split into ten equal groups. The first group was left as the test set, and others were used to train the model. Then the second group was left as the test set, and so on. In the end, we will have ten models with performance evaluation. The final result was expressed as the weighted average outcome from these ten models.

  1. In the introduction I suggest authors to further describe the recent application of machine learning in clinics also besides trauma, as for example for COVID-19 or Alzheimer's disease prediction to highlight the benefit of applying ML in the hospital to predict mortality and disease progression (doi: 10.2459/JCM.0000000000001329; doi: 10.3390/jpm11010014.)

Reply: Thank you for the constructive comments, we revised our introduction and added the recent application of machine learning.

Introduction paragraph 3 line:

 Recently, increasing studies have collected ICU patients’ data to predict clinical events or outcomes. [6,7,10,13–16]. Vezzoli, Marika et al. has applied machine learning for  prediction of in-hospital mortality in coronavirus disease[24] and except for prediction, Abate, Giulia et al. used machine learning to identified Alzheimer Disease in preclinical and prodromal Stages[25].As the available dataset increased and the technique improved, ML showed earlier advantages in detecting high-risk events earlier.[12]

  1. I suggest including the Yuden index in ROC curves and report the 95 % CI of AUC.

Reply:  Thank you for the constructive comments. We include the Yuden index in ROC curves and the 95 % CI of AUC in our result part. We also modified figure 3 according to your suggestion.

Page 3

2.5. Statistical Analysis

Demographic and outcomes data are presented as medians with interquartile ranges. For continuous variables, extensive exploratory data analysis, including calculation of maximum, minimum, median, and mean value, was done, and the frequency of the test, and the difference between each test were recorded. The accuracy, sensitivity, specificity, and area under the receiver operating characteristic curve (AUC) with a 95% confidence interval (CI) were used to assess the model performance.

Page 7

3.3. The Performance of Prediction Models

Two machine learning algorithms were applied in our study. Logistic model tree(LMT) achieve 73.8% precision, 71.3% sensitivity, 75.9% specificity at the best Youden index of 0.486, with an AUC of 0.816 (95% CI: 0.798-0.834) for the validation set. The Random Forest algorithm achieves 73.6% precision, 73.6% sensitivity, and 73.7% specificity at the best Youden index of 0.479, with an AUC of 0.809 (95% CI: 0.791-0.827) for the validation set (Table 3). The ROC Curve is shown in Figure 1.

  1. Is it possible to calculate the de-long test between the two ROCs to demonstrate if there is a difference between the two algorithms used (Logistic model tree and RF: Random Forest) .

Reply: Thank you for your thoughtful comments. We have tried to use the de-long test for these two ROCs. However, we found that the ROC result of these two modules is not simply used prediction value to calculate a single dataset but uses two class groups (class positive and class negative, as decision tree method ) ROC then gets the weighted average value in 10-fold cross-validation. Therefore, the de-long test may not be applicable to demonstrate the difference between these two ROC.

  1. It is not clear to the reader why the need of two different ML-algorithm. none outperformed the others. Authors should better explain the rationale used in ML-method selection. 

Reply: Thank you for your important comments. We have tried many different ML-algorithm, and we found these two ML-algorithm showed the best accuracy. Therefore, we want to express the concept that these two algorithms may be good candidates for data mining and prediction of ICU data. We didn’t only choose one because this is an exploratory research, and the data we input to the algorithm is still not perfect. Therefore, though these two algorithms showed no obvious difference, the trend may change in the future if we keep inputting further data, even from other centers. That is the reason that we not only chose the best-performed algorithm in our study. We have additionally described the concept in the limitation part.

Page 7.

  1. Discussion

As our research presented, we have built an efficient and feasible algorithm to achieve an appropriate level of prediction function. The current study is an exploratory research. Therefore, we only use the ten-fold cross-validation method to validate our result without an external validation dataset. Further external validation will be conducted for the best algorithm chosen.

Reviewer 2 Report

Thank you for the opportunity to review a retrospective observational study regarding the machine learning-based analysis model for prediction of hemorrhage in ICU patients after trauma. The authors aimed to develop two prediction models using logistic model tree and random forest algorithms. They found those models achieved an area under the curve of 0.816 and 0.809, respectively. These prediction models may be valuable for predicting delayed warning risk in trauma patients. However, I have some concerns about this paper. See the following issues.

Major issue:

1.     Needs validation study

Although study participants were relatively small (n=2,218), validation study for those prediction models is essential. Therefore, I would suggest that the authors should divide participants into two groups for analysis: development and validation groups. If possible, sensitivity analysis is also required.

2.     Surgical treatments

Initial surgical treatments except TEA before ICU admission were not considered as an analytic variable. Why didn’t the authors enter those variables into ML models?

3.     Mean value imputation

For missing data, the authors imputed it with the mean value of the whole database. However, this approach can lead to biased estimates of statics. In general, multiple imputation is a popular approach for addressing the presence of missing data. I would suggest the authors should use multiple imputation for missing data.

Minor issue:

1.     Transfusion

Did the authors consider the volume of transfusion before/after ICU admission?

Author Response

Thank you for reviewing our manuscript and giving us constructive suggestions. Below we respond point-by-point to your recommendations.

  1. Needs validation study:
    Although study participants were relatively small (n=2,218), validation study for those prediction models is essential. Therefore, I would suggest that the authors should divide participants into two groups for analysis: development and validation groups. If possible, sensitivity analysis is also required.

Reply: Thank you for your constructive suggestion. In this manuscript. We demonstrated the result as an exploratory study. External validation is crucial, as you suggested before the algorithm can be applied to the clinical environment. We will split the data as you suggest in the future study for a single algorithm experiment. The study is ongoing in our institute currently. We have emphasized this part in the discussion. In the current study, we used the 10-fold cross-validation method to validate our model. The whole dataset was divided into 10 groups, 9 for model development and 1 as a test group repeated this process 10 times every time and had one outcome, and in the end, we got a weighted average outcome from these 10 results. We used this method to decrease the bias that may be created during divided group training and testing for only one time.

Page 7.

  1. Discussion

As our research presented, we have built an efficient and feasible algorithm to achieve an appropriate level of prediction function. The current study is an exploratory research. Therefore, we only use the ten-fold cross-validation method to validate our result without an external validation dataset. Further external validation will be conducted for the best algorithm chosen.

  1. Surgical treatments:
    Initial surgical treatments except TEA before ICU admission were not considered as an analytic variable. Why didn’t the authors enter those variables into ML models?

Reply: Thank you for your thoughtful opinion. This is a crucial point. Limited by our EHR system, we can only get the gross surgery category by surgical code and can’t clearly identify these operation’s indications. For example, exploratory laparotomy may be indicated for acute bleeding by liver laceration but also may be done due to hollow organ perforation that is not necessarily related to bleeding. The only way to do this is to read the operation record to make sure the intervention is related to bleeding. That detailed information is not available in both CGRD or the trauma registry. Therefore, instead of including these interventions without clearly identifying the purpose of these interventions, which may be misleading, we didn't put these interventions in our analysis.

But as you said, these variables are actually important for our population. What we did was initially build an essential model for trauma patients at ICU, as we mentioned in the limitation paragraph. In the future, we will try to get more detailed EHR records and better selection method and build a better and better model based on our initial model.

We added the limitation of surgical intervention interpretations in the limitation part as you suggested.

  1. Limitation

Interventions like transarterial embolization or exploratory laparotomy/thoracotomy also point to bleeding events that can’t be controlled by conservative treatment. In the databases we used in the current study, we only have raw codes rather than details of procedures. Further work to identify these events and add to our model or build models by the separate event is necessary and helpful for physicians’ decision-making.

  1. Mean value imputation:
    For missing data, the authors imputed it with the mean value of the whole database. However, this approach can lead to biased estimates of statics. In general, multiple imputation is a popular approach for addressing the presence of missing data. I would suggest the authors should use multiple imputations for missing data.

Reply: Thank you for the great suggestion. We used the mean value imputation to simplify the imputation process to make the module not overfit to other parameters. As you suggested, we tried multiple imputations to generate another result but found the performance is slightly improved but not significantly better than our current results. The weighted ROC of random forest and LMT model is 0.817(95% CI: 0.803-0.831) and 0.827(95% CI: 0.814-0.840).

Therefore, we decided to report the result still using mean value imputation in the manuscript. We add the reason in the limitation part.

  1. Limitation

The reason that we can’t afford too much missing data in variables is to ensure our outcome relates more to the real clinical situations. We also chose the mean value imputation instead of other complicated imputation methods to avoid over-manipulating the data.

  1. Transfusion
    Did the authors consider the volume of transfusion before/after ICU admission?

Reply: Thank you for your constructive suggestion. We tried to record blood transfusion, but different from the general EHR of CGRD, the most detailed record of blood volume cost for patients is in the blood bank, which is not included in the CGRD dataset we applied for. Though there was also a blood transfusion order recorded at EHR, while we rechecked the data, we found multiple mismatches of blood transfusion time/volume while comparing to the nursing record and physician’s order(sometimes the order was canceled by oral order or text order that not recorded at EHR). Therefore, we can’t put transfusion as a variable this time. We are working on a blood bank record application for further study. We have added this problem to the limitation.

  1. Limitation.

Third, as mentioned before, we haven’t included all bleeding-related events in our study, so we could miss some patients with delayed bleeding but not expressed as Hb dropping. The blood transfusion event time was also not included in our study due to inaccurate data in the current database.

Round 2

Reviewer 1 Report

Authors adequately reply to most of the comments and suggestions requested by reviewers. The manuscript has been improved and now it is suitable for publishing. 

Author Response

Thank you for the great suggestions and for allowing us to improve our manuscript.

Reviewer 2 Report

Thank you for the revision. I have no more suggestions.

Author Response

(The authors gave the same response as above.)
